# Differences in HIV-1 reservoir size, landscape characteristics, and decay dynamics in acute and chronic treated HIV-1 Clade C infection

Kavidha Reddy[1], Guinevere Q Lee[2], Nicole Reddy[1,3], Tatenda JB Chikowore[1,4], Kathy Baisley[1,5], Krista L Dong[6,7,8], Bruce D Walker[6,7,8], Xu G Yu[6,8], Mathias Lichterfeld[6,8,9], Thumbi Ndung'u[1,3,4,6,7]*

[1]Africa Health Research Institute, Durban, South Africa; [2]Weill Cornell Medical College, New York, United States; [3]University of KwaZulu-Natal, Durban, South Africa; [4]University College London, London, United Kingdom; [5]London School of Hygiene and Tropical Medicine, London, United Kingdom; [6]Ragon Institute of MGH, MIT and Harvard, Cambridge, United States; [7]HIV Pathogenesis Programme (HPP), The Doris Duke Medical Research Institute, University of KwaZulu-Natal, Durban, South Africa; [8]Harvard Medical School, Boston, United States; [9]Brigham and Women's Hospital, Boston, United States

*For correspondence: thumbi.ndungu@ahri.org

## eLife Assessment

This **fundamental**, clearly written, and timely manuscript links the timing of ART with the kinetics of total and intact proviral HIV DNA. The conclusions are interesting and novel, and the importance of the work is high because the focus is on African women and clade C virus, both of which are under-studied in the HIV reservoir field. The strength of the evidence is **compelling**. Overall, this work will be of very high interest to scientists and clinicians in the HIV cure/persistence fields.

**Abstract** Persisting HIV reservoir viruses in resting CD4 T cells and other cellular subsets are a barrier to cure efforts. Early antiretroviral therapy (ART) enables post-treatment viral control in some cases, but mechanisms remain unclear. We hypothesised that ART initiated before peak viremia impacts HIV-1 subtype C reservoirs. We studied 35 women at high risk of infection from Durban, South Africa, identified with hyperacute HIV by twice-weekly HIV-RNA testing. Participants included 11 starting ART at a median of 456 (297–1203) days post-onset of viremia (DPOV) and 24 at 1 (1–3) DPOV. Peripheral blood mononuclear cells (PBMCs) were used to measured total HIV-1 DNA by droplet digital PCR (ddPCR) and sequence viral reservoir genomes by full-length proviral sequencing (FLIP-seq). ART during hyperacute infection blunted peak viremia (p<0.0001), but contemporaneous total HIV-1 DNA did not differ (p=0.104). Over 1 year, a decline of total HIV-1 DNA was observed in early treated persons (p=0.0004), but not late treated. Among 697 viral genome sequences, the proviral genetic landscape differed between untreated, late treated, and early treated groups. Intact genomes after 1 year were higher in untreated (31%) versus late treated (14%) and early treated (0%). Treatment in both late and early infection caused more rapid decay of intact (13% and 51% per month) versus defective (2% and 35%) viral genomes. However, intact genomes persisted 1 year post chronic treatment but were undetectable with early ART. Early ART also reduced phylogenetic diversity of intact genomes and limited cytotoxic T lymphocyte immune escape variants in the reservoir. Overall, ART initiated in hyperacute HIV-1 subtype C infection did not impact reservoir seeding

but was associated with rapid intact viral genome decay, reduced genetic complexity, and limited immune escape, which may accelerate reservoir clearance in combination with other interventional strategies.

## Introduction

Although antiretroviral therapy (ART) effectively controls HIV, it does not eradicate persisting virus from resting CD4 T cells and other cellular subsets. ART is therefore lifelong, with many side effects and may be financially unsustainable particularly in low- to middle-income countries that are most affected by the HIV epidemic. A cure is therefore necessary but difficult to achieve as the persistent HIV reservoir, which is established soon after infection (*Siliciano et al., 2003*; *Finzi et al., 1999*; *Wong et al., 1997*), hinders these efforts through its complex nature (*Deeks et al., 2021*).

The HIV-1 reservoir is heterogenous and dynamic in genotypic, phenotypic, and cellular composition and also differs between individuals and populations, making the source of persistent infection difficult to target. The lability of the viral reservoir has been demonstrated by evidence of clonal expansion that occurs by homeostatic proliferation (*Vandergeeten et al., 2013*; *Chomont et al., 2009*), antigenic stimulation (*Mendoza et al., 2020*), or integration into genic regions controlling cell growth (*Jiang et al., 2020*; *Maldarelli et al., 2014*; *Wagner et al., 2014*). Additionally, its size and genetic composition changes over time with intact and defective proviral genomes displaying differences in their rate of decay during suppressive ART (*White et al., 2022b*; *Massanella et al., 2021*; *Peluso et al., 2020*). The heterogeneity of the HIV-1 reservoir landscape is further obscured by the integration patterns of virus-intact genomes that determine their ability to generate replication competent viral particles when stimulated (*Jiang et al., 2020*). Moreover, recent studies show that the viral reservoir is not restricted to CD4 T cells but persists in other cell types such as monocytes, macrophages (*Andrade et al., 2020*; *Venanzi Rullo et al., 2019*; *Hiener et al., 2017*; *Honeycutt et al., 2016*; *Koppensteiner et al., 2012*; *Wightman et al., 2010*), and follicular dendritic cells (*Banga et al., 2023*; *Olivetta et al., 2020*; *Ollerton et al., 2020*; *Keele et al., 2008*) with each having distinct biological characteristics.

Among the many studies directed towards HIV-1 cure, some have demonstrated that early ART initiation may be a crucial step in limiting the viral reservoir and achieving post-treatment viral control (*Namazi et al., 2018*; *Sáez-Cirión et al., 2013*). Even though early treatment does not prevent reservoir establishment, it does result in a rapid decline of viremia and substantially reduces HIV reservoir size (*Massanella et al., 2021*; *Ananworanich et al., 2016*; *Buzon et al., 2014*; *Ananworanich et al., 2012*) which delays viral load rebound and results in sustained virological remission following discontinuation of ART (*Li et al., 2016*; *Williams et al., 2014*; *Sáez-Cirión et al., 2013*). Other described benefits of starting therapy soon after primary infection include preservation of immune function and restriction of viral diversification (*Jeewanraj et al., 2023*; *Naidoo et al., 2024*; *Naidoo et al., 2021*; *Muema et al., 2020*; *Lee et al., 2019*; *Ndhlovu et al., 2019*; *Mabuka et al., 2017*; *Ananworanich et al., 2016*; *Kløverpris et al., 2016*; *Buzon et al., 2014*; *Moir et al., 2010*; *Rosenberg et al., 2000*; *Rosenberg et al., 1997*). However, there is a paucity of data that describes the impact of the timing of treatment initiation on the characteristics of the reservoir.

Most studies describing characteristics of the HIV reservoir are from men living with HIV-1 subtype B in the Global North (*White et al., 2022b*; *Peluso et al., 2020*; *Hiener et al., 2017*; *Bruner et al., 2016*) and show that irrespective of treatment timing, intact proviral genomes decay more rapidly compared to defective viral genomes and that defective viral genomes accumulate quickly during acute infection and are predominant within the viral reservoir.

HIV-1 reservoir studies in African populations are limited, particularly in subtype C HIV-1 infection, which is the most prevalent form of HIV-1 globally and predominates in southern Africa. The epidemic in Africa is characterised by extensive viral diversity with multiple subtypes, human genetic heterogeneity which influences immunological and disease outcomes, and unique co-morbidities that modulate HIV reservoirs and immune responses (*Joussef-Piña et al., 2022*; *Sarabia and Bosque, 2019*; *Nedelec et al., 2016*; *Messele et al., 1999*). The design of a globally applicable HIV cure strategies and interventions to target the viral reservoir depend on a deeper understanding of the variability in the size, composition, and characteristics of the genetic landscapes of persisting reservoir genomes in African populations with non-subtype B HIV infections. Moreover, data are lacking on reservoir

**Table 1.** Characteristics of study participants.

| Characteristics | Chronic treated (n=11) | Acute treated (n=24) |
| --- | --- | --- |
| Age (years) | 21 (19–24) | 21 (18–24) |
| Sex | | |
| Female, n (%) | 11 (100%) | 24 (100%) |
| Male, n (%) | 0 (0%) | 0 (0%) |
| Race/ethnicity, n (%) | | |
| Black | 11 (100%) | 24 (100%) |
| Fiebig stage I at detection, n (%) | 10 (91%) | 21 (88%) |
| Treatment initiation (DPOV) | 456 (297–1203) | 1 (1-3) |
| Time to suppression (days) | 104 (30–215) | 16 (6–116) |
| CD4 nadir (cells/µl) | 383 (204–502) | 561 (258–859) |
| CD4 pre-infection (cells/µl) | 991 (395–1377) | 872 (573–1612) |
| CD4 at study enrollment (baseline) (cells/µl) | 716 (204–1377) | 863 (421–2075) |
| Peak plasma viral load (log copies/ml) | 7.04 (5.89–7.80) | 4.21 (2–7.30) |
| *Protective HLA allele, n (%) | 6 (55%) | 11 (46%) |
| Treatment regimen containing, n (%) | | |
| FDC | 11 (100%) | 24 (100%) |
| Raltegravir | 0 (0%) | 16 (67%) |

*HLA-B74:01, HLA-B57:02, HLA-B57:03, HLA-B58:01, HLA-B81:01.

characteristics in women, despite known sex differences in immune responses and in viral load during primary infection that could potentially impact the reservoir (*Chang et al., 2013*; *Meier et al., 2009*; *Sterling et al., 2001*).

In this study we performed an extensive longitudinal analysis of HIV-1 subtype C proviral characteristics, that would be informative for understanding mechanisms of reservoir establishment, in a unique hyperacute infection cohort in Durban, South Africa (*Dong et al., 2018*; *Ndung'u et al., 2018*). The cohort was designed to identify acute infection before peak viremia (Fiebig stages I to III) providing HIV-1 testing twice a week to young women at high risk for HIV-1 infection in a region with high population prevalence (*Dong et al., 2018*). Following changes in treatment guidelines in South Africa that allowed for ART initiation regardless of CD4 counts, all study participants were offered ART, including those who were newly detected with acute infection who received ART on average a day after first detection of plasma viremia. Study participants underwent frequent clinical follow-up and sampling following infection and initiation of ART, allowing us to study HIV reservoir establishment and proviral evolution from the earliest possible stages of infection. We hypothesised that the timing of ART will impact HIV-1 proviral genome characteristics in terms of the size, genetic composition, and decay dynamics. The findings of this study provide insights into HIV-1 proviral characteristics that could inform viral targeting strategies for reservoir control in African populations.

## Results

### Total proviral DNA load kinetics following early and late treatment

In this analysis we included 35 participants (*Supplementary file 1*), of whom 11 first initiated treatment during chronic infection at a median of 456 days (297–1203) post detection of viremia and 24 who were treated during acute infection at a median of 1 day (1–3) post detection of viremia. All participants were female and 31 (89%) were identified with acute infection at Fiebig stage I. Additional participant characteristics are shown in *Table 1*.

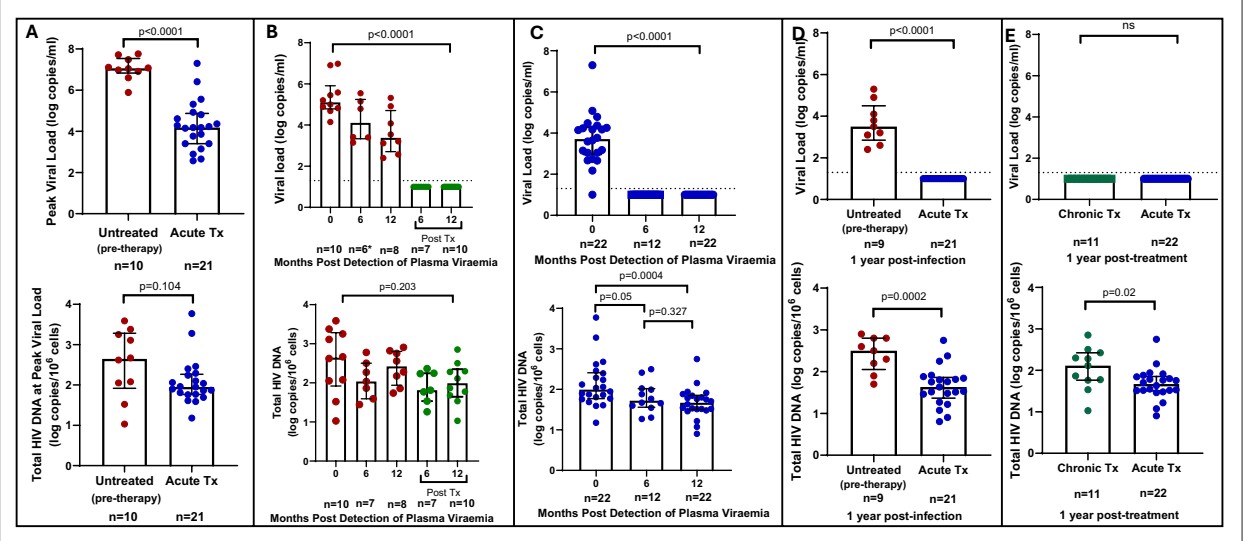

**Figure 1.** Plasma viral load and total HIV DNA in acute treated and chronic treated individuals. (**A**) Peak viral load (parametric t-test) and total HIV DNA (non-parametric t-test) measured at peak viral load in untreated (pre-therapy) and acute treated individuals. (**B**) Longitudinal viral load (Kruskal-Wallis ANOVA) (*1 viral load measurement was unavailable) and total HIV DNA (Kruskal-Wallis ANOVA) in untreated acute infection and after 6 and 12 months of treatment. (**C**) Longitudinal viral load (Kruskal-Wallis ANOVA) and total HIV DNA (non-parametric t-tests) in acute treated individuals. (**D**) Viral load and total HIV DNA (parametric t-test) after 1 year of treatment in chronic and acute treated individuals. Median and interquartile range (error bars) are represented.

The online version of this article includes the following source data for figure 1:

**Source data 1.** Droplet digital PCR (ddPCR) numerical data used to generate *Figure 1*.

We first quantified total HIV-1 DNA which incorporates all forms of intracellular HIV-1 DNA, both intact and defective, including integrated and unintegrated forms, as well as linear and circularised 2-LTR and 1-LTR forms. Total HIV-1 DNA measurements were performed longitudinally, from baseline (1–3 days following detection of HIV), at the time of peak viral load and was also assessed at 6 and 12 months' post infection for untreated participants and 6 and 12 months' post-treatment initiation for late and early treated participants. As expected, treatment during acute infection resulted in a significantly reduced peak plasma viral load (median = 4.18 log copies/ml, IQR, 3.40–4.87) compared to untreated acute infection (median = 7.06 log copies/ml, IQR, 6.83–7.54) (p<0.0001, *Figure 1A*, top panel). However, at time of peak viremia, the untreated and the early treated groups did not differ in total proviral DNA load (*Figure 1A*, bottom panel). Longitudinal measurements showed that treatment initiated during chronic infection resulted in a significant decline in plasma viral load to undetectable levels after 1 year (p<0.0001, *Figure 1B*, top panel), however it did not reduce total proviral load (*Figure 1B*, bottom panel). In contrast, treatment initiated during acute infection resulted in both a rapid decrease of plasma viremia so that all participants had undetectable viremia at 1-year post ART (p<0.0001, *Figure 1C*, top panel) and steady decrease of total proviral load over the same time period (p=0.0004) (*Figure 1C*, bottom panel). Even though treatment initiation during both chronic and acute infection resulted in complete suppression of plasma viral load after 1 year (*Figure 1D*, top panel), total proviral load was still detectable with the early treated group having 1.3 times lower levels of total proviral HIV DNA compared to the chronic treated group (p=0.02, *Figure 1D*, bottom panel). These results indicate that early treatment leads to a measurable decline in proviral DNA during the first year of treatment that is not seen when therapy is initiated during chronic infection.

## Factors associated with total proviral DNA load after 1 year of suppressive ART

To further understand the impact of host, virological, and immunological factors, as well as timing of treatment on the establishment and maintenance of the HIV reservoir, we analysed the associations between virological and immunological markers of clinical disease progression and HIV-1 proviral DNA after 1 year of treatment (*Table 2*). Analyses for each treatment group were performed independently

**Table 2.** Multivariate analysis of factors that predict total HIV-1 proviral DNA load after 1 year of treatment.

| Stage at treatment initiation | Variables | Co-efficient | Standard error | t | p-Value | p-Value summary | 95% confidence interval |
|---|---|---|---|---|---|---|---|
| Acute infection | Nadir CD4 | –0.0007424 | 0.00102 | 0.7277 | 0.4773 | ns | –0.002905–0.001420 |
| | Pre-infection CD4 | –0.0001861 | 0.0004233 | 0.4395 | 0.6661 | ns | –0.001083–0.0007113 |
| | Baseline* CD4 | 0.0002574 | 0.0003727 | 0.6906 | 0.4997 | ns | –0.0005326–0.001047 |
| | Peak VL | 0.1972 | 0.07938 | 2.485 | 0.0244 | * | 0.02895–0.3655 |
| Chronic infection | Nadir CD4 | –0.006633 | 0.0007088 | 9.358 | <0.0001 | **** | –0.008367 to –0.004898 |
| | Pre-infection CD4 | 0.0002514 | 0.0003896 | 0.6454 | 0.5425 | ns | –0.0007019–0.001205 |
| | Baseline* CD4 | 0.001501 | 0.0002905 | 5.166 | 0.0021 | ** | 0.0007899–0.002211 |
| | Peak VL | 0.2658 | 0.09698 | 2.741 | 0.0337 | * | 0.02848–0.5031 |

*At study enrolment.

using multivariate regression models with HIV-1 DNA levels as the dependent variable and other factors, specifically nadir CD4, pre-infection CD4, baseline CD4 counts, and peak viral load, as the independent predictor variables. The analysis showed that when treatment was initiated during acute infection, only peak plasma viral load was significantly associated with levels of HIV-1 proviral DNA after 1 year of treatment (p=0.02). However, when treatment was initiated during chronic infection, both baseline CD4 count (measured 1–3 days after detection of HIV) (p=0.002) and peak plasma viral load (p=0.03) positively associated with HIV-1 proviral DNA levels, while there was a significant inverse association with nadir CD4 (p<0.0001). Other factors such as total viral burden (area under the viral load curves), CD4:CD8 ratio at enrolment, protective human leukocyte antigen (HLA) alleles, and type of treatment regimen were not associated with HIV-1 proviral DNA measured after 1 year of treatment (data not shown). These data indicate that both host and viral characteristics impact the establishment and maintenance of the viral reservoir.

## Longitudinal genotypic characterisation of HIV-1 DNA

Quantification of total HIV-1 DNA by ddPCR as described above is based on the amplification of a short 127 base pair fragment of the HIV-1 genome, and thus detects defective viruses that are incapable of replication, thereby overestimating the size and functionality of the reservoir. To address this, we next performed single template near full-genome sequencing to determine potential replication competency by establishing the distribution of genome-intact and genome-defective latent viruses within cells. Viral genome intactness was determined by the HIVSeqinR v2.7.1 computational bioinformatics pipeline (*Lee et al., 2019*).

For this analysis we studied 24 participants: The chronic infection (late treatment) group (n=11) consisted of individuals who remained untreated for over 1 year following infection and before treatment initiation. Longitudinal sampling at untreated time points was available for nine of these individuals whereas in two individuals, samples were only available post treatment initiation. The acutely treated group (n=13) received treatment 1–3 days post detection (*Figure 2A*). We generated a total of 697 sequences (GenBank accession numbers OR991333-OR991737 and MK643536-MK643827) after sampling a median of 1.4 million peripheral blood mononuclear cells (PBMCs) (0.02–4.3 million) per sampling time point. Genome-intact viruses (*Figure 2B*) accounted for 35% (247/697) of the total pool and were detected in 23 participants (12 from the early treatment group and 11 from the late treatment group), with a median of 8 genome-intact viruses (range = 1–60) per study participant. Phylogenetic analysis revealed a significant difference in the mean pairwise distances of intact viral sequences derived from early treated (median = 0.12%, IQR, 0.07–0.21) compared to late treated participants

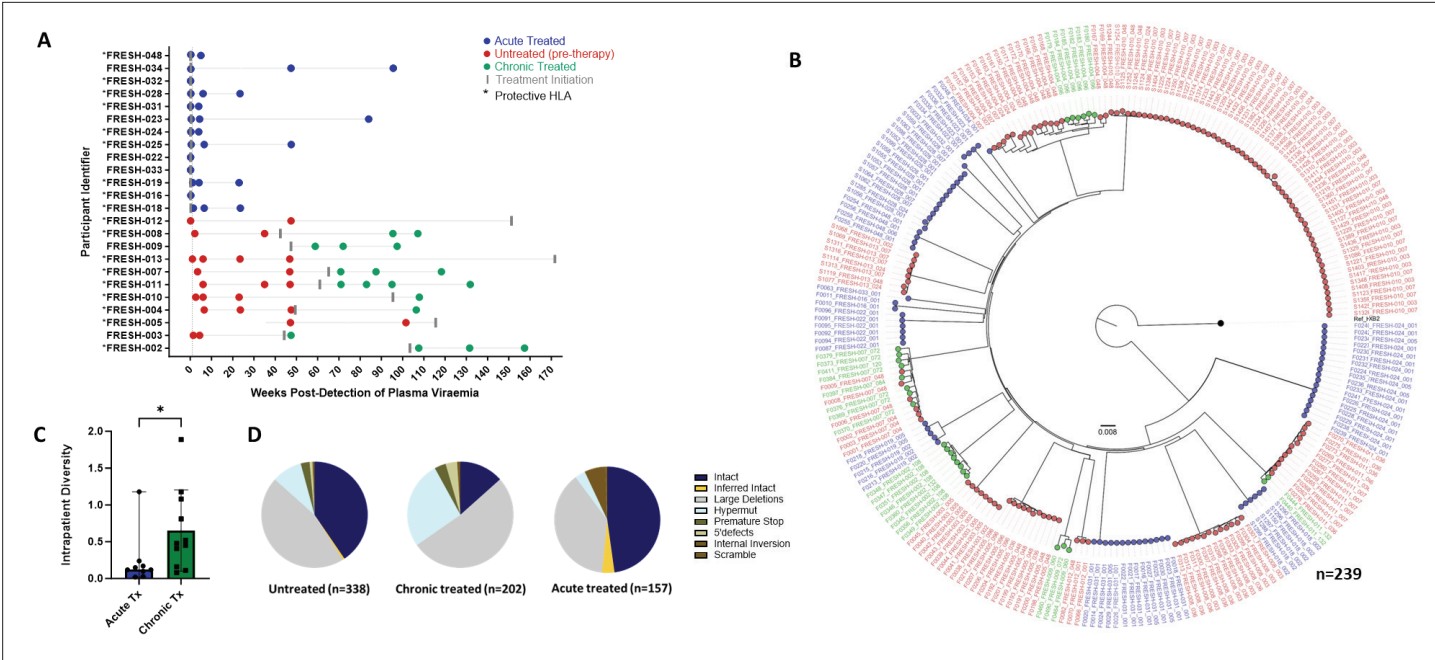

**Figure 2.** Genotypic characterisation of HIV-DNA sequences. (**A**) Peripheral blood mononuclear cell (PBMC) sequencing time points in untreated (red), chronic treated (green), and early treated (blue) study participants where each dot represents a sampling time point. Time of treatment initiation is shown by the vertical grey bar. (**B**) Approximately maximum-likelihood phylogenetic tree of intact HIV-1 DNA genomes constructed using FastTree2. This method was chosen to resolve full-viral-genome sequences with extreme homology; branch lengths were likely inflated. Viral genomes derived from acute treated participants are marked with (*). (**C**) Comparison of intraparticipant mean pairwise distances between early and late treated participants. (**D**) Spectrum of HIV genome sequences detected during untreated acute infection, late treated chronic infection, and acute treated infection.

The online version of this article includes the following figure supplement(s) for figure 2:

**Figure supplement 1.** In this cohort of HIV-1 subtype C, genome deletions were most frequently observed between *integrase* and *envelope* relative to Gag (p<0.0001–0.001).

(median = 0.48%, IQR, 0.16–1.08) (p=0.04) (*Figure 2C*). Overall, 56% of the intact genomes collected in this study were obtained from the untreated study arm, while 11% were obtained from late treated chronic infection and 33% from acutely treated infections (*Figure 2D*).

Longitudinal studies revealed that defective proviruses accumulated rapidly during the course of HIV-1 infection with a relative contribution of 65% (450/697) to the total pool of proviral genomes detected. The majority of defective viral genomes collected in this study were detected during untreated infection (44% (199/450), left-most pie), while 39% (175/450) were detected during late treated chronic infection and 17% (76/450) during acute treated infection. Defective genomes contributed 59% to the proviral population in untreated infection, 87% in late (chronic) treated, and 48% in acutely treated infection (*Figure 2D*). Overall, defective genomes also accumulated quickly after onset of infection and were detectable at a proportion of 47% within the first month irrespective of treatment status (data not shown). Large internal deletions within viral genomes were the most common defect with relative frequencies of 77%, 60%, and 79% in untreated infection, chronic treated infection, and acute treated infection respectively among the pool of defective genomes. Overall, these gene deletions occurred significantly more frequently between *integrase* and *envelope* in the *integrase* to *envelope* gene segment compared to *gag* (p<0.0001–0.001), with *nef* being similar to *gag* (*Figure 2—figure supplement 1*). APOBEC-induced hypermutations were the second most common defect observed in untreated (16%) and late (chronic) treated (31%) infection. However, in acute treated infection, hypermutations were relatively infrequent, comprising only 7% of the genome-defective pool. Premature stop codons in one of *gag*, *pol*, or *env* occurred at a frequency of 5%, 4%, and 13% as a percentage of defective genomes in sequences from untreated, chronic, and acute treated infections respectively. Internal inversions (1%, 1%, 1%), and 5' psi defects (2%, 4%, 0%) were other types of genome defects that were detected at minor frequencies in untreated, chronically treated, and acutely treated infections respectively.

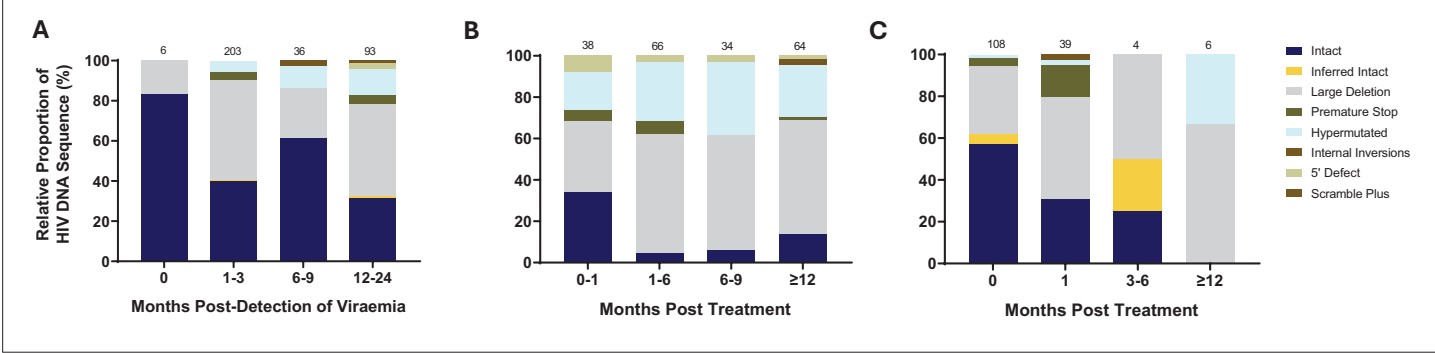

**Figure 3.** Evolution of the proviral genetic landscape. Relative proportions of intact and defective viral genomes measured longitudinally in (**A**) untreated acute infection for 2 years, (**B**) late (chronic) treated infection for 1 year, and (**C**) early (acute) treated infection for 1 year. The number of genomes sampled at each time point is indicated above each vertical bar.

The online version of this article includes the following figure supplement(s) for figure 3:

**Figure supplement 1.** Clonal expansion of infected cells was detected in both defective (orange) and intact (blue) genomes in late and early treated study participants.

To further understand the impact of ART timing on the composition, evolution, and dynamics of the HIV-1 proviral landscape over time, we next performed a stratified analysis of the relative proportions of viral genome sequences in each study arm over 1 year of follow-up (*Figure 3A–C*). Genome-intact viruses were detectable throughout the course of untreated infection while genome-defective viruses also accumulated over this period (*Figure 3A*). Initiation of ART during chronic infection, at a median of 456 days after detection of plasma viremia, resulted in a decrease in the relative proportion of genome-intact viruses over 1 year of treatment (34–14%). However, genome-intact viruses were not completely eradicated and were easily detectable after 12 months of treatment (*Figure 3B*). Additionally, genomes with large deletions and hypermutations became more prominent in the chronic treated group over 1 year of treatment. In contrast, there was a more rapid decrease in the proportion of genome-intact viruses following ART initiation in acute infection such that these viruses were no longer detectable at our sampling depth after 1 year of treatment (57% to 0%) (*Figure 3C*). Hypermutated viruses were also less prominent before 12 months during early treated infection (*Figure 3C*). These data suggest that early treatment initiation facilitates faster clearance of genome-intact viruses in the blood compared to late treatment.

## Contribution of clonal expansion to maintenance of proviral populations

Studies show that more than 50% of the latent HIV reservoir is maintained by clonal expansion (*Liu et al., 2020*). We assessed viral genome sequences to determine the extent of persistence of infected cell clones after primary infection. Viral genome sequences sharing 100% identity by FLIP-seq was used as a marker of clonal expansion of infected cells as previous studies have shown that proviral genomes that were 100% identical share the same viral integration site whereas proviruses with different integration sites do not share 100% sequence identity (*Einkauf et al., 2019*). At our sampling depth, we detected clonal expansion in 3/11 (27%) participants who were treated during chronic infection and 4/13 (30%) participants treated during acute infection, showing that in subtype C infection clonal expansion of infected cells occurred as early as 1 day post detectable viremia (*Figure 3—figure supplement 1*). Defective clones were detected in two late treated participants at proportions of 6% and 13% of total proviral population, while intact clones were identified in one late treated participant at a proportion of 6% of the total proviral pool. In contrast, a higher proportion of intact clones were detected in early treated participants at 33%, 30%, 37%, and 24% of the total proviral pool. Although the data is limited and needs to be interpreted with caution, this suggests that clonal expansion of intact proviral genomes is more likely to occur when treatment is initiated early, likely due to the early inhibition of viral replication that prevents the accumulation and seeding of defective viral genomes into the viral reservoir.

## Decay kinetics of intact and defective proviruses

Studies show that the biology and decay dynamics of genome-intact viruses within the viral reservoir likely differ from that of the genome-defective provirus pool (*White et al., 2022b*; *Peluso et al., 2020*). However, the effect of ART timing on the rate of decay of these different pools of viruses is not well known and has not been investigated in African populations where immune responses and viral genetic heterogeneity may well result in population-specific differences. Here, we observed that the absolute proportions of both genome-intact and -defective viruses per million PBMCs sampled decreased in both early treated and late treated participants over the 1-year follow-up period of this study (*Figure 4A and B*).

To estimate potential differences in the rate of change between genome-intact and -defective viruses within each treatment group, we used a linear mixed effects regression model with random intercepts to account for the correlation between repeated observations from the same individual. We fit a model with log DNA copies as the response variable, and time, treatment group, and a time-group interaction as fixed effects, with participant as a random effect. The analysis was restricted to the first 6 months after starting ART as regular measurements for both groups were available over this period.

Among the acute treated, genome-intact proviruses decreased by 0.308 log copies per month in the first 6 months after starting ART, corresponding to a decline of 51% per month (p<0.001, *Figure 4C*). In contrast, among the chronic treated, intact proviruses decreased by only 0.059 log copies per month, corresponding to a decline of 13% per month; however, this decrease was not statistically significant (p=0.68) (*Figure 4E*).

Genome-defective proviruses also decreased significantly in the acute treated group by 0.190 log copies per month in the first 6 months after starting ART, corresponding to a decline of 35% (p=0.01). However, in the chronic treated group the change in the number of log copies of defective provirus in the first 6 months was only 0.015 (p=0.88) corresponding to a decline of just 3.4% per month (*Figure 4D and E*). These results indicate that early treatment is associated with a faster decline of both genome-intact and -defective proviruses compared to late treatment.

## CTL epitope diversity in the latent reservoir

The emergence of escape mutations in viral epitopes as a mechanism to evade HLA class I-restricted immune responses, specifically of CD8+ cytotoxic T lymphocytes (CTL), drives viral diversification and is a significant challenge in developing effective therapies against HIV (*Goonetilleke et al., 2009*; *Borrow et al., 1997*; *Goulder et al., 1997*; *Koup and Ho, 1994*). We investigated the impact of late compared to early ART initiation on CTL epitope diversity and escape in the HIV proviral genomes by longitudinally analysing Gag, Nef, and Pol CTL epitopes, (*Mamrosh et al., 2022*) from single genome viral sequences (excluding only hypermutated sequences), that are restricted by HLA genotypes B*57:02, B*57:03, B*58:01, B*81:01, and A*74:01. These HLA genotypes have been associated with protection against disease progression in HIV-1 subtype C infection (*Goulder and Walker, 2012*). CTL epitope mutations were classified according to the Los Alamos HIV Molecular Immunology Database (*Mamrosh et al., 2022*). Protective HLA genotypes were present in 7/11 (64%) late treated participants and in 7/13 (54%) early treated participants. In the presence of relevant restricting HLA genotypes, mutations compared to the Clade C consensus were detected in 12% of participants with Gag, 23% with Pol, and 27% with Nef targeted epitopes after 1 year of follow-up when treatment was initiated late (*Figure 5A–C*) in contrast to 0%, 0%, and 8% respectively when treatment was initiated early (*Figure 5G–I*), suggesting that chronic treatment is associated with the retention of a wide spectrum of CTL escape mutations within proviral genomes compared to early treatment. Escape mutations detected at baseline (up to 1 month after infection) in the presence of restricting HLA genotypes were present in 3% of participants within Gag, 19% within Pol, and 23% (*Figure 5A–C*) within Nef targeted epitopes when treatment was initiated later compared to and 0%, 13%, and 11% respectively (*Figure 5G–I*) with early treatment. Escape mutations observed in early treated participants were present in the earliest sequences that were derived close to the time of infection and therefore likely represent transmitted escape variants. Similar proportions of transmitted escape mutations were present in participants who did not have a protective HLA genotype and remained unchanged after 1 year (*Figure 5D–F and J–L*).

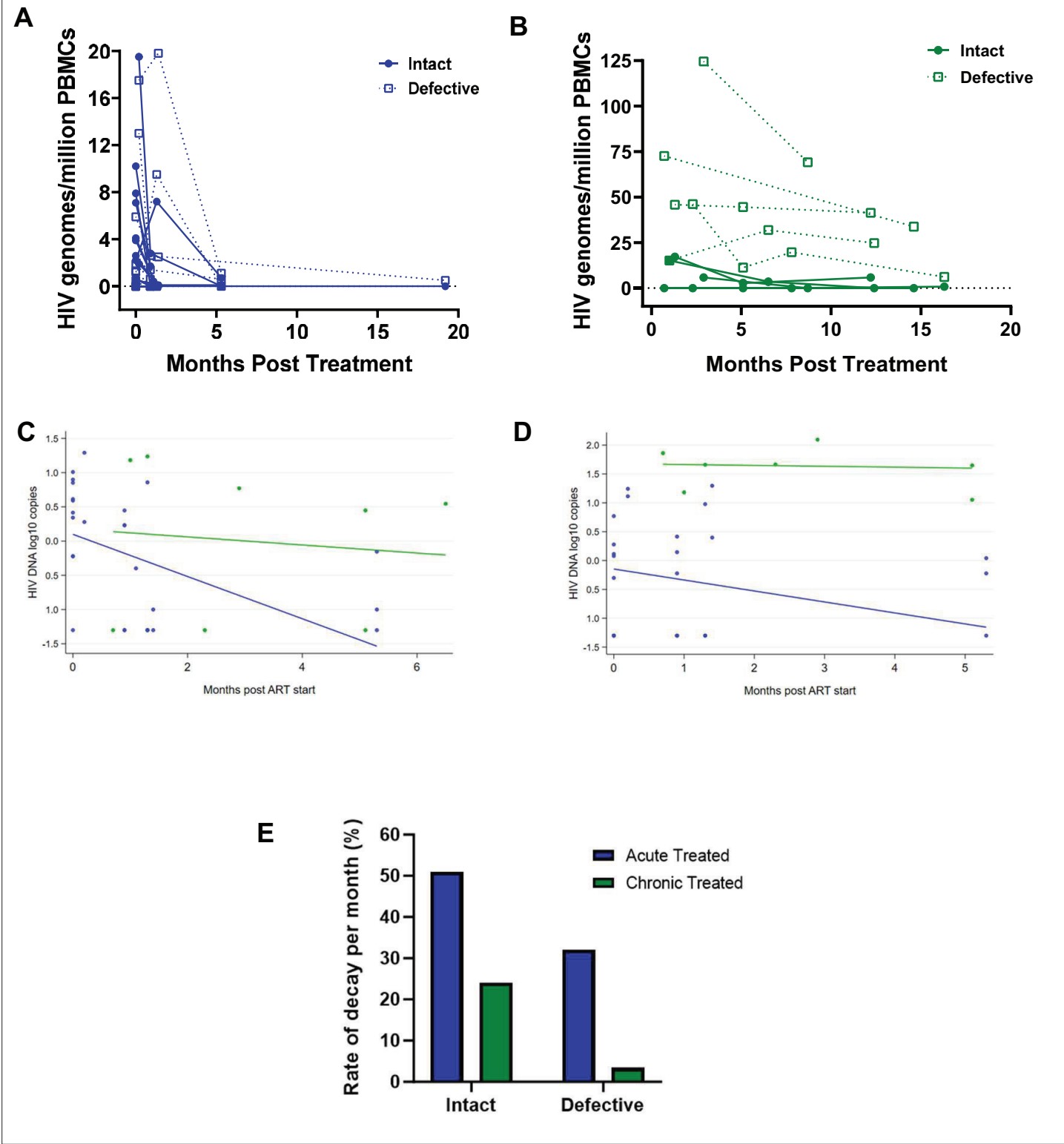

**Figure 4.** Decay kinetics of intact and defective proviruses. Absolute frequencies of intact and defective HIV-1 DNA sequences per million peripheral blood mononuclear cells (PBMCs) during the first year of infection following treatment during (**A**) acute infection and (**B**) chronic infection. Longitudinal analysis of the change in (**C**) intact and (**D**) defective provirus copies in the 6 months after antiretroviral therapy (ART) initiation, comparing the acute treated (blue) and chronic treated (green) groups. Dots represent a measurement from a given participant; solid lines are slopes estimated from linear mixed effect model. (**E**) Comparison of the monthly rate of decay of intact and defective proviruses in acute and chronic treated infection.

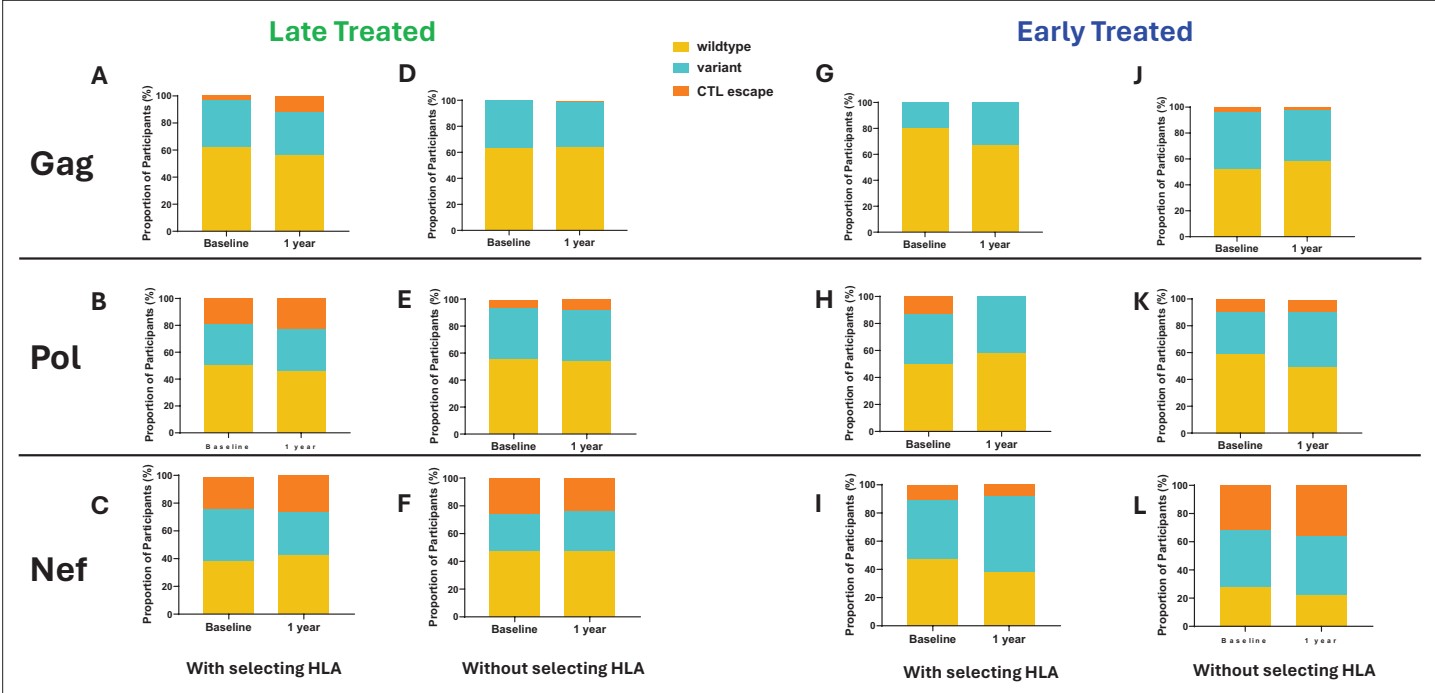

**Figure 5.** Comparison of cytotoxic T lymphocytes (CTL) epitope diversity in late compared to early treated participants. Proportion of participants with wildtype, variant, and CTL escape at baseline (within 1 month of infection) and up to 1 year of infection in Gag (**A, D, G, J**), Pol (**B, E, H, K**), and Nef (**C, F, I, L**) epitopes in participants with protective human leukocyte antigen (HLA) genotypes (**A, B, C, G, H, I**) and without protective HLA genotypes (**D, E, F, J, K, L**).

## Discussion

In this study we used a well-characterised acute HIV infection longitudinal cohort of untreated, early treated and late treated study participants, to perform an extensive quantitative and qualitative analysis of the HIV-1 subtype C proviral landscape. Our aim was to determine whether the timing of treatment has an impact on the viral reservoir size, genetic landscape, and decay kinetics. This was a longitudinal study where we measured total proviral load levels by ddPCR and characterised the genetic landscape of the proviral genomes by next-generation sequencing using FLIP-seq. Total HIV-1 DNA is an important biomarker of clinical outcomes (*Rouzioux and Avettand-Fenoël, 2018*; *Rouzioux et al., 2018*; *Avettand-Fènoël et al., 2016*; *Williams et al., 2014*). We found that HIV DNA was detectable at high levels during primary infection and even in participants who were treated during acute infection, total HIV DNA levels measured at peak viremia were very similar to untreated participants. In contrast, we found that early but not late treatment was associated with steady decline of total proviral load over the first year of ART. These observations confirm previous data that the viral reservoir is seeded at the earliest stages of infection, possibly before peak viremia (*Colby et al., 2018*; *Bruner et al., 2016*; *Whitney et al., 2014*; *Siliciano et al., 2003*; *Chun et al., 1997*). However, in contrast, studies have suggested that early ART intervention restricts the seeding of the HIV reservoir in long-lived central memory CD4 T cells (*Ananworanich et al., 2012*). Moreover, it has recently been demonstrated that a small fraction of deeply latent genetically intact proviruses are archived in CD4 T cells during the very first weeks of infection (*Gantner et al., 2023*). However, even though early ART initiation has been associated with continued HIV DNA reduction during long-term ART after 10 years of follow-up (*Buzon et al., 2014*), it remains detectable in most individuals indicating that early ART alone is insufficient to achieve viral eradication.

We also examined the association of clinical and virological factors with the levels of proviral DNA after 1 year of treatment. In both early and late treatment groups, peak plasma viral load was associated with HIV DNA levels. Moreover, in the late treatment group there was a positive association with CD4 at enrolment (baseline) and an inverse correlation with nadir CD4 counts. Considering that the majority of CD4 T cells remain uninfected it is likely that this does not represent a higher number

of target cells, and this warrants further investigation. Similar associations with nadir CD4 have been reported previously (*Fourati et al., 2014*; *Boulassel et al., 2012*; *Burgard et al., 2009*), suggesting that during untreated progressive HIV infection, ongoing viral replication may drive the accumulation of long-lived latently infected cells that repopulate the immune system by expansion during successful ART.

Even though total HIV DNA has been shown to be a clinically significant marker of the HIV reservoir, it does not distinguish between replication-competent and -defective viruses that contribute to the viral reservoir. The unique design of the FRESH cohort based on frequent HIV screening and sampling intervals of high-risk uninfected participants allowed us to examine the dynamics of the proviral landscape from the earliest stages of infection (Fiebig I) up to 1 year of ART by near-full-length viral genome sequencing. We performed a comparison of the proviral populations between study participants who were treated during the acute phase of infection and those initiating ART during chronic infection. Defective genomes accumulate rapidly after the onset of infection and contributed to almost half of the proviral population within the first 4 weeks of infection irrespective of treatment status. Consistent with other studies (*Lee et al., 2019*; *Pinzone et al., 2019*; *Bruner et al., 2016*) the overall proviral landscape in the untreated and late treated participants was dominated by defective viruses suggesting that prolonged ongoing viral replication before treatment initiation leads to the accumulation of defective viral genomes. However, we observed fewer defective genomes in the FRESH cohort with acute ART initiation in comparison to other studies (*Hiener et al., 2017*; *Bruner et al., 2016*). These differences could be attributed to the timing of treatment initiation where in the aforementioned studies early treatment ranged from 100 days to 0.6–3.4 months after infection respectively. This interesting observation may indicate that a more developed antiviral immune responses in the other studies may have led to more defective genomes compared to FRESH following ART initiation. However, this will require additional studies.

A further analysis of the composition of defective viral genomes revealed that genome deletions were most frequently observed between *integrase* and *envelope*. A previous study showed that large deletions are non-random and occur at hotspots in the HIV-1 genome with *envelope* being a hotspot for large deletions (*Bruner et al., 2019*; *Bruner et al., 2016*). Additionally, we noted differences in the frequencies of hypermutations compared to other studies in subtype B cohorts, suggesting that the timing of ART initiation and sex- or race-based differences in immunological factors that impact the reservoir may play a role (*Bruner et al., 2016*).

Genome-intact viruses were easily detectable throughout untreated infection but decreased after treatment. In participants who were treated during chronic infection, genome-intact viruses were still detectable after 1 year of ART compared to early treated individuals where they were no longer detectable. With our limited sampling size and depth, of a median of 1.4 million PBMC (0.02–4.3 million) per sampling time point, we cannot rule out that intact genomes may be retrieved with further sampling and investigation into tissue reservoirs of the participants who initiate ART during acute infection (*Reeves et al., 2018*). However, these findings do provide further evidence that introducing ART during acute HIV infection limits the size of the HIV reservoir considerably compared to treatment during chronic infection (*Massanella et al., 2021*; *Namazi et al., 2018*; *Ananworanich et al., 2016*; *Buzon et al., 2014*; *Sáez-Cirión et al., 2013*; *Ananworanich et al., 2012*).

The intact proviral DNA assay which is a more scalable method that uses multiplexed ddPCR to measure individual proviruses and differentiates intact from defective proviruses without the need for long-distance PCR has been suggested to provide more accurate quantitative information about the size and composition of the latent reservoir compared to near-full-genome sequence methods (*White et al., 2022a*). However, this assay would also be affected by sampling depth and was only recently developed and optimised for quantification of subtype C HIV (*Buchholtz et al., 2024*).

Studies have shown that during suppressive ART, intact and defective proviruses have different rates of decay that occurs in a biphasic manner (*White et al., 2022b*; *Massanella et al., 2021*; *Peluso et al., 2020*). Our analysis of decay kinetics was limited to a linear mixed effects regression model as we were unable to fit a model for biphasic decay which requires frequent proviral DNA measurements. Our analysis was further restricted to the first 6 months of ART as viral genomes were difficult to detect by FLIP-seq in early treated participants after this time. We found that indeed intact genomes decay faster than defective genomes in both early and late treatment groups. Cells containing intact viral genomes likely represent productively infected cells that may be preferentially targeted for

clearance by the host immune response or eliminated by viral cytopathic effects (*White et al., 2022b*; *Peluso et al., 2020*). Moreover, early treatment results in a faster decline of both intact and defective genomes compared to treatment initiated during chronic infection and is suggestive of a more effective immune clearance mechanism from preserved immune function. Despite this, it is estimated that 226 years of effective ART is necessary to decrease intact proviral DNA levels by 4 $\log_{10}$ (*Peluso et al., 2020*). This further indicates that early ART in combination with novel interventional strategies will be needed to achieve a faster viral eradication.

Several studies show that clonal expansion of HIV-infected cells plays an important role in maintaining the HIV reservoir further contributing to the challenge of eradicating HIV (*Coffin et al., 2019*; *Einkauf et al., 2019*; *Cohn et al., 2015*; *Ho et al., 2013*). Our findings suggest that clonal expansion of intact viral genomes detected predominantly in early treated infection may contribute to the maintenance of the HIV reservoir in these study participants. Further studies that extend beyond 1 year of treatment will help elucidate whether these clones expand further after several years of treatment. Our analysis of CTL epitope escape mutations, known to drive viral diversification, revealed that early treatment minimises the emergence of CTL escape in Gag, Pol, and Nef epitopes despite these participants having protective HLA alleles. In contrast, CTL escape was detected when treatment was initiated during chronic infection specifically in the well-characterised TW10 gag epitope restricted by HLA B57/58. Transmitted CTL escape mutations detected within the first few weeks of infection were common in both treatment groups confirming previous data from this study population (*Gounder et al., 2015*). Moreover, the rapid rate of clearance of viral genomes observed with early treatment could be attributed to the lower proportion of CTL escape mutations in the proviral genomes of these participants compared with those treated later.

To our knowledge this is the first study in an African population, dominated by subtype C HIV infection, that examined the impact of the timing of ART initiation on HIV reservoir establishment in a longitudinal setting. Moreover, our data focused on women who are underrepresented in reservoir and cure studies globally. Our data showed that early ART initiation does not blunt proviral DNA seeding in immune reservoirs, but it nevertheless results in a more rapid decay of intact viral genomes, decreases genetic complexity and immune escape. Although early ART alone may not be sufficient to eradicate the persisting viral reservoir, our results suggest that when combined with interventional strategies, it is more likely to achieve an effective HIV cure.

## Materials and methods

**Key resources table**

| Reagent type (species) or resource | Designation | Source or reference | Identifiers | Additional information |
|---|---|---|---|---|
| Cell line (*Homo sapiens*, human) | 8E5 LAV cell line (CEM) | NIH HIV Reagent Program | CAT #95, RRID:CVCL_3484 | Served as a positive control for viral genome sequencing |
| Biological sample (*Homo sapiens*, human) | Human PBMCs | *Dong et al., 2018* | FRESH Cohort | Ethics Approval Reference Numbers: BF131/11 and 2012-P001812 |
| Sequence-based reagent | LTR-gag Forward primer | *Lee et al., 2019* | ddPCR Primer | TCTCGACGCAGGACTCG |
| Sequence-based reagent | LTR-gag Reverse primer | *Lee et al., 2019* | ddPCR Primer | TACTGA CGCTCTCGCACC |
| Sequence-based reagent | LTR-gag probe | *Lee et al., 2019* | ddPCR Probe | /56- FAM/CTCTCTCCT/ZEN/TCTAGCCTC/ 31ABkFQ/ |
| Sequence-based reagent | RPP30 forward primer | *Lee et al., 2019* | ddPCR Primer | GATTTGGACCTGC GAGCG |
| Sequence-based reagent | RPP30 reverse primer | *Lee et al., 2019* | ddPCR Primer | GCGGCTGTCTCCACAAGT |
| Sequence-based reagent | RPP30 probe | *Lee et al., 2019* | ddPCR Probe | /56- FAM/ CTGACCTGA/ZEN/AGGCTCT/31ABkFQ/ |

*Continued on next page*

*Continued*

| Reagent type (species) or resource | Designation | Source or reference | Identifiers | Additional information |
|---|---|---|---|---|
| Sequence-based reagent | U5-623F1 | *Lee et al., 2019* PCR Primer | | AAATCTCTAGCAGTGGCGCCCGAACAG |
| Sequence-based reagent | U5-638F2 | *Lee et al., 2019* PCR Primer | | GCGCCCGAACAGGGACYTGAAARCGAAAG |
| Sequence-based reagent | U5-547R2 | *Lee et al., 2019* PCR Primer | | GCACTCAAGGCAAGCTTTATTGAGGCTTA |
| Sequence-based reagent | U5-601R1 | *Lee et al., 2019* PCR Primer | | TGAGGGATCTCTAGTTACCAGAGTC |
| Commercial assay or kit | ddPCR supermix No dTUPs | Bio-Rad | SCR_026079 CAT #1863023 | |
| Commercial assay or kit | ddPCR droplet generator oil | Bio-Rad | SCR_026081 CAT #BBRD1863004 | |
| Commercial assay or kit | ddPCR droplet reader oil | Bio-Rad | SCR_026084 CAT #BBRD1864110 | |
| Commercial assay or kit | DNeasy Blood and Tissue extraction kit | QIAGEN | SCR_026085 CAT #69506 | |
| Commercial assay or kit | Bio-Rad QX200 AutoDG Droplet Digital PCR System | Bio-Rad | RRID:SCR_019714 | |
| Commercial assay or kit | ddPCR supermix No dTUPs | Bio-Rad | SCR_026079 CAT #1863023 | |
| Commercial assay or kit | Platinum Taq DNA Polymerase High Fidelity | Invitrogen | CAT # 11304102 | |
| Software, algorithm | QX Manager Standard edition version 1.2 | Bio-Rad | SCR_026078 | |
| Software, algorithm | HIVSeqinR v2.7.1 | *Lee et al., 2019* | Bioinformatics Pipeline | |
| Software, algorithm | GraphPad Prism v10 | GraphPad Software Inc | Graphs and Statistics | |

## Study design and participants

This was a longitudinal study of the Females Rising through Education, Support, and Health (FRESH) cohort, a prospective, observational study of 18- to 23-year-old HIV uninfected women at high risk for HIV acquisition, established in Umlazi, Durban, South Africa (*Dong et al., 2018*; *Ndung'u et al., 2018*). Finger prick blood draws were collected from FRESH study participants twice a week and subjected to HIV-1 RNA testing, with the aim of detecting acute HIV infection during Fiebig stage I. The study included a socioeconomic intervention program and HIV prevention interventions including PrEP that coincided with study visits to address challenges faced by the young women that likely contribute to the increased risk of HIV acquisition in this setting. If a participant acquired HIV-1 infection while on the study, blood samples were collected weekly for a month, then monthly until 3 months' post infection, then monthly for 1 year and every 3 months thereafter. Days post onset of viremia (DPOV) was calculated as the interval between the first positive HIV test and the date of sample collection. Unique participant identifier numbers were assigned to the participants and are only known to the research group.

Study participants recruited during the first 19 months of the study did not receive antiretroviral treatment immediately after detection of acute HIV infection but were monitored and referred for treatment when they became eligible according to national treatment guidelines at the time. The South African national treatment eligibility criteria subsequently changed allowing the immediate initiation of ART for all people living with HIV, including those with acute HIV infection as recommended under the World Health Organization's universal test and treat policy (*WHO, 2015*). The treatment

schedule was a three-drug daily oral regimen of 300 mg tenofovir disoproxil fumarate, 200 mg emtricitabine, and 600 mg efavirenz. Additionally, following the change in South African first-line treatment guidelines to include an integrase inhibitor, raltegravir (400 mg twice a day) was introduced as a fourth drug and was continued for 90 days after viral suppression (<20 copies/ml).

For this study participants were categorised into three groups (untreated, late (chronic) treated, and early (acute) treated) where, 11 remained untreated during acute infection and later started ART during chronic infection at a median of 456 (297–1203) DPOV, while 24 started ART at a median of 1 (1–3) DPOV. Participants were studied at 0, 1, 3, 6, 9, 12, and 24 months post onset of viremia and up to 12 months post treatment. Peak viraemia refers to the highest recorded viral load in all participants.

## Quantification of total HIV-1 DNA

Measurement of total HIV-1 DNA was performed as previously described (*Lee et al., 2017*). Total DNA was extracted from total PBMC samples using DNeasy Blood & Tissue Kits (QIAGEN). ddPCR (Bio-Rad) was used to measure total HIV-1 DNA and host cell concentration with primers and probes covering HIV-1 5′ LTR-*gag* HXB2 coordinates 684–810 (forward primer 5′-TCTCGACG CAGGACTCG-3′, reverse primer 5′-TACTGACGCTCTCGCACC-3′ probe/56-FAM/CTCTCTCCT/ ZEN/TCTAGCCTC/ 31ABkFQ/, and human RPP30 gene38 forward primer 5′-GATTTGGACCTG CGAGCG-3′, reverse primer 5′-GCGGCTGTCTCCACAAGT-3′, probe/56-FAM/CTGACCTGA/ZEN/ AGGCTCT/31AbkFQ/). Thermocycling conditions for ddPCR were: 95°C for 10 min, 45 cycles of 94°C for 30 s and 60°C for 1 min, 72°C for 1 min. Thereafter droplets from each sample were analysed on the Bio-Rad QX200 Droplet Reader and data were analysed using QuantaSoft software (Bio-Rad).

## Single genome amplification and deep sequencing of near-full-length HIV-1 DNA

Near-full-length proviral sequences were generated as previously described (*Lee et al., 2017*). Total HIV-DNA copy number as determined by ddPCR was used to calculate the DNA sample dilution to achieve one PCR-positive reaction in every three reactions. This method of limiting dilution gives a Poisson probability of 85.7% that each PCR amplicon originated from a single HIV-DNA template (*Lee, 2021*). A nested PCR approach was used to amplify the near-full genome using one unit of Platinum Taq DNA Polymerase High Fidelity (Invitrogen) with the following primers: first round PCR: forward primer 5′-AAATCTCTAGCAGTGGCGCCCGAACAG-3′, reverse primer 5′-TGAGGGATCTCTAGTT ACCAGAGTC-3′; second round PCR: forward primer 5′-GCGCCCGAACAGGGACYTGAAARCGAAA G-3′, reverse primer 5′-GCACTCAAGGCAAGCTT TATTGAGGCTTA-3′ (HXB2 coordinates 638–9632, 8994 bp). The 20 μl reaction mix contained 1× reaction buffer, 2 mM MgSO₄, 0.2 mM dNTP, 0.4 μM each of forward and reverse primers. The thermal cycling conditions were 2 min at 92°C, 10 cycles (10 s at 92°C, 30 s at 60°C, 10 min at 68°C), 20 cycles (10 s at 92°C, 30 s at 55°C, 10 min at 68°C), 10 min at 68°C, 4°C infinite hold.

## Illumina MiSeq and bioinformatics analysis

All PCR amplicons detectable by gel electrophoresis were subjected to Illumina MiSeq sequencing and thereafter the resulting small reads were de novo assembled using in-house UltraCycler v1.0 (Brian Seed and Huajun Wang, unpublished) (*Lee et al., 2017*). Viral genome intactness was inferred by the computational bioinformatics pipeline HIVSeqinR v2.7.1 (*Lee et al., 2019*).

## Cell line

DNA isolated from the 8E5/LAV (derivative of CEM), human-derived cell line (NIH AIDS Reagent Program, Catalogue #95), which has a single copy of integrated HIV-1 genome, served as a positive control for viral genome sequencing. The identity of the cell line was authenticated by sequencing the proviral genome using the single copy viral DNA genome amplification assay, FLIP-seq, which showed 100% sequence identity in all positive controls which were included on every sequencing plate. Since mycoplasma is not infectable by HIV, mycoplasma contamination was not controlled for in our experiment.

## HLA typing

HLA typing was performed using a targeted next-generation sequencing method as previously described (*Lin et al., 2023*).

## Statistical methods

GraphPad Prism 10 was used to perform summary statistical analyses and comparisons among study groups using Fishers' exact, Mann-Whitney and Kruskal-Wallis, and multiple linear regression analysis.

## Acknowledgements

The study cohort and sample collection were supported in part by grants from the Bill and Melinda Gates Foundation (OPP1066973 and OPP1146433), Gilead Sciences, Inc (Grant ID #00406), the International AIDS Vaccine Initiative (IAVI) (UKZNRSA1001), the NIAID (R37AI067073), the Witten Family Foundation, the Dan and Marjorie Sullivan Foundation, the Mark and Lisa Schwartz Foundation, Ursula Brunner, the AIDS Healthcare Foundation, and the Harvard University Center for AIDS Research (CFAR, P30 AI060354, which is supported by the following institutes and centers co-funded by and participating with the US National Institutes of Health: NIAID, NCI, NICHD, NHLBI, NIDA, NIMH, NIA, FIC, and OAR). Raltegravir used for immediate treatment was donated by Merck & Co., Inc.This work was also partially supported through the Sub-Saharan African Network for TB/HIV Research Excellence (SANTHE) which is funded by the Science for Africa Foundation to the Developing Excellence in Leadership, Training and Science in Africa (DELTAS Africa) programme (Del-22–007) with support from Wellcome Trust and the UK Foreign, Commonwealth & Development Office and is part of the EDCPT2 programme supported by the European Union; the Bill & Melinda Gates Foundation (INV-033558); and Gilead Sciences Inc (19275). All content contained within is that of the authors and does not necessarily reflect positions or policies of any SANTHE funder. For the purpose of open access, the author has applied a CC BY public copyright licence to any Author Accepted Manuscript version arising from this submission. The authors thank all participants in the FRESH cohort who have made this study possible. The authors thank the Massachusetts General Hospital Center for Computational & Integrative Biology DNA Core, specifically Dr. Nicole Stange-Thomann, Dr. Amy Avery, Ms. Kristina Belanger, and Mr. Huajun Wang, for providing them with the Illumina MiSeq deep sequencing service used in this manuscript. The 8E5 LAV cell line was obtained through the NIH HIV Reagent Program, Division of AIDS, NIAID, NIH: Human Immunodeficiency Virus 1 (HIV-1) Lymphadenopathy-Associated Virus (LAV)-Infected 8E5 Cells, ARP-95, contributed by Dr. Thomas Folks.

## Additional information

### Funding

| Funder | Grant reference number | Author |
|---|---|---|
| Bill and Melinda Gates Foundation | OPP1066973 | Thumbi Ndung'u Bruce D Walker |
| Gilead Sciences | Grant ID #00406 | Thumbi Ndung'u |
| International AIDS Vaccine Initiative | UKZNRSA1001 | Thumbi Ndung'u |
| Bill and Melinda Gates Foundation | OPP1146433 | Thumbi Ndung'u Bruce D Walker |
| National Institute of Allergy and Infectious Diseases | R37AI067073 | Bruce D Walker |
| Harvard University Center for AIDS Research | P30 AI060354 | Bruce D Walker |
| Wellcome Trust | Del-22–007 | Thumbi Ndung'u |
| Bill & Melinda Gates Foundation | INV-033558 | Thumbi Ndung'u |

| Funder | Grant reference number | Author |
| --- | --- | --- |
| Gilead Sciences Inc | Grant ID #19275 | Thumbi Ndung'u |

The funders had no role in study design, data collection and interpretation, or the decision to submit the work for publication. For the purpose of Open Access, the authors have applied a CC BY public copyright license to any Author Accepted Manuscript version arising from this submission.

## Author contributions

Kavidha Reddy, Conceptualization, Data curation, Formal analysis, Investigation, Methodology, Writing – original draft, Writing – review and editing; Guinevere Q Lee, Conceptualization, Data curation, Software, Formal analysis, Writing – review and editing; Nicole Reddy, Tatenda JB Chikowore, Investigation, Methodology; Kathy Baisley, Formal analysis, Writing – review and editing; Krista L Dong, Project administration, Writing – review and editing, Samples and clinical/demographical data collection; Bruce D Walker, Funding acquisition, Writing – review and editing, Samples and clinical/demographical data collection; Xu G Yu, Mathias Lichterfeld, Conceptualization, Writing – review and editing; Thumbi Ndung'u, Conceptualization, Formal analysis, Supervision, Funding acquisition, Writing – review and editing

## Author ORCIDs

Kavidha Reddy ⓘ https://orcid.org/0000-0002-7024-0574
Mathias Lichterfeld ⓘ https://orcid.org/0000-0001-9865-8350
Thumbi Ndung'u ⓘ https://orcid.org/0000-0003-2962-3992

## Ethics

The Biomedical Research Ethics Committee of the University of KwaZulu-Natal (BF131/11) and the Institutional Review Board of Massachusetts General Hospital (2012-P001812) approved the study. All participants provided written informed consent.

Reviewer #1 (Public review): https://doi.org/10.7554/eLife.96617.4.sa1
Reviewer #2 (Public review): https://doi.org/10.7554/eLife.96617.4.sa2
Author response https://doi.org/10.7554/eLife.96617.4.sa3

# Additional files

## Supplementary files

Supplementary file 1. Clinical and biological characteristics of 35 study participants. *Deleterious human leukocyte antigen (HLA) class I alleles (red), **protective HLA class I alleles (green).

MDAR checklist

## Data availability

Sequencing Data has been deposited in GenBank under accession codes OR991333 to OR991737 and MK643536 to MK643827. ddPCR data analysed for this study are included in the supporting files; Source data 1 contains the ddPCR data used to generate *Figure 1—source data 1*.

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
