## [Editor Report · eLife Assessment]

This **fundamental**, clearly written, and timely manuscript links the timing of ART with the kinetics of total and intact proviral HIV DNA. The conclusions are interesting and novel, and the importance of the work is high because the focus is on African women and clade C virus, both of which are understudied in the HIV reservoir field. The strength of the evidence is **compelling**. Overall, this work will be of very high interest to scientists and clinicians in the HIV cure/persistence fields.

---

## [Referee Report · Reviewer #1 (Public review)]

The authors sought to determine the impact of early antiretroviral treatment on the size, composition, and decay of the HIV latent reservoir. This reservoir represents the source of viral rebound upon treatment interruption and therefore constitutes the greatest challenge to achieving an HIV cure. A particular strength of this study is that it reports on reservoir characteristics in African women, a significantly understudied population, of whom some have initiated treatment within days of acute HIV diagnosis. With the use of highly sensitive and current technologies, including digital droplet PCR and near full-length genome next-generation sequencing, the authors generated a valuable dataset for investigation of proviral dynamics in women initiating early treatment compared to those initiating treatment in chronic infection. The authors confirm previous reports that early antiretroviral treatment restricts reservoir size, but further show that this restriction extends to defective viral genomes, where late treatment initiation was associated with a greater frequency of defective genomes. Furthermore, an additional strength of this study is the longitudinal comparison of viral dynamics post-treatment, wherein early treatment was shown to be associated with a more rapid rate of decay in proviral genomes, regardless of intactness, over a period of one year post-treatment. While it is indicated that intact genomes were not detected after one year following early treatment initiation, sampling depth is noted as a limitation of the study by the authors, and caution should thus be taken with interpretation where sequence numbers are low. Defective genomes are more abundant than intact genomes and are therefore more likely to be sampled. Early treatment was also associated with reduced proviral diversity and fewer instances of polymorphisms associated with cytotoxic T-lymphocyte immune selection. This is expected given that rapid evolution and extensive immune selection are synonymous with HIV infection in the absence of treatment, yet points to an additional benefit of early treatment in the context of immune therapies to restrict the reservoir.

This is one of the first studies to report the mapping of longitudinal intactness of proviral genomes in the globally dominant subtype C. The data and findings from this study therefore represent a much-needed resource in furthering our understanding of HIV persistence and informing broadly impactful cure strategies. The analysis on clonal expansion of proviral genomes may be limited by higher sequence homogeneity in hyperacute infection i.e., cells with different proviral integration sites may have a higher likelihood of containing identical genomes compared to chronic infection.

Overall, these data demonstrate the distinct benefits of early treatment initiation at reducing the barrier to a functional cure for HIV, not only by restricting viral abundance and diversity but also potentially through the preservation of immune function and limiting immune escape. It therefore provides clues to curative strategies even in settings where early diagnosis and treatment may be unlikely.

---

## [Referee Report · Reviewer #2 (Public review)]

HIV infection is characterized by viral integration into permissive host cells - an event that occurs very early in viral-host encounter. This constitutes the HIV proviral reservoir and is a feature of HIV infection that provides the greatest challenge for eradicating HIV-1 infection once an individual is infected.

This study looks at how starting HIV treatment very early after infection, which substantially reduces the peak viral load detectable (compared to untreated infection), affects the amount and characteristics of the viral reservoir. The authors studied 35 women in South Africa who were at high risk of getting HIV. Some of these women started HIV treatment very soon after getting infected, while others started later. This study is well-designed and has as its focus a very well characterized cohort. Comparison groups are appropriately selected to address proviral DNA characterization and dynamics in the context of acute and chronic treated HIV-1. The amount of HIV and various characteristics of the genetic makeup of the virus (intact/defective proviral genome) was evaluated over one year of treatment. Methods employed for proviral DNA characterization are state-of-the-art and provide in-depth insights into the reservoir in peripheral blood.

While starting treatment early didn't reduce the amount of HIV DNA at the outset, it did lead to a gradual decrease in total HIV DNA quantity over time. In contrast, those who started treatment later didn't see much change in this parameter. Starting treatment early led toa faster decrease in intact provirus (a measure of replication-competence), compared to starting treatment later. Additionally, early treatment reduced genetic diversity of the viral DNA and resulted in fewer immune escape variants within intact genomes. This suggests that collectively having a smaller intact replication-competent reservoir, less viral variability, and less opportunity for virus to evade the immune system - are all features that are likely to facilitate more effective clearance of viral reservoir, especially when combined with other intervention strategies.

Major strengths of the study include the cohort of very early treated persons with HIV and the depth of study. These are important findings, particularly as the study was conducted in HIV-1 subtype C infected women (more cure studies have focussed on men and with subtype B infection)- and in populations most affected by HIV and in need of HIV cure interventions. This is highly relevant because it cannot be assumed that any interventions employed for reducing/clearing the HIV reservoir would perform similarly in men and women or across different populations. Other factors also deserve consideration and include age, and environment (e.g. other comorbidities and coinfections).

---

## [Author Response]

The following is the authors’ response to the previous reviews.

**Public Reviews:**

**Reviewer #1:**
(1) Given that this is one of the first studies to report the mapping of longitudinal intactness of proviral genomes in the globally dominant subtype C, the manuscript would benefit from placing these findings in the context of what has been reported in other populations, for example, how decay rates of intact and defective genomes compare with that of other subtypes where known.

Most published studies are from men living with HIV-1 subtype B and the studies are not from the hyperacute infection phase and therefore a direct head-to-head comparison with the FRESH study is difficult. However, we can cite/highlight and contrast our study with a few a few examples from acute infection studies as follows.

a. Peluso et. al., JCI, 2020, showed that in Caucasian men (SCOPE study), with subtype B infection, initiating ART during chronic infection virus intact genomes decayed at a rate of 15.7% per year, while defective genomes decayed at a rate of 4% per year. In our study we showed that in chronic treated participants genomes decreased at a rate of 25% (intact) and 3% (defective) per month for the first 6 months of treatment.

b. White et. al., PNAS, 2021, demonstrated that in a cohort of African, white and mixed-race American men treated during acute infection, the rate of decay of intact viral genomes in the first phase of decay was <0.3 logs copies in the first 2-3 weeks following ART initiation. In the FRESH cohort our data from acute treated participants shows a comparable decay rate of 0.31 log copies per month for virus intact genomes.

c. A study in Thailand (Leyre et. al., 2020, Science Translational Medicine), of predominantly HIV-1 CRF01-AE subtype compared HIV-reservoir levels in participants starting ART at the earliest stages of acute HIV infection (in the RV254/SEARCH 010 cohort) and participants initiating ART during chronic infection (in SEARCH 011 and RV304/SEARCH 013 cohorts). In keeping with our study, they showed that the frequency of infected cells with integrated HIV DNA remained stable in participants who initiated ART during chronic infection, while there was a sharp decay in these infected cells in all acutely treated individuals during the first 12 weeks of therapy. Rates of decay were not provided and therefore a direct comparison with our data from the FRESH cohort is not possible.

d. A study by Bruner et. al., Nat. Med. 2016, described the composition of proviral populations in acute treated (within 100 days) and chronic treated (>180 days), predominantly male subtype B cohort. In comparison to the FRESH chronic treated group, they showed that in chronic treated infection 98% (87% in FRESH) of viral genomes were defective, 80% (60% in FRESH) had large internal deletions and 14% (31% in FRESH) were hypermutated. In acute treated 93% (48% in FRESH) were defective and 35% (7% in FRESH) were hypermutated. The differences frequency of hypermutations could be explained by the differences in timing of infection specifically in the acute treated groups where FRESH participants initiate ART at a median of 1 day after infection. It is also possible that sex- or race-based differences in immunological factors that impact the reservoir may play a role.

This study also showed that large deletions are non-random and occur at hotspots in the HIV-1 genome. The design of the subtype B IPDA assay (Bruner et. al., Nature, 2019) is based on optimal discrimination between intact and deleted sequences - obtained with a 5′ amplicon in the Ψ region and a 3′ amplicon in Envelope. This suggest that Envelope is a hotspot for large while deletions in Ψ is the site of frequent small deletions and is included in larger 5′ deletions. In the FRESH cohort of HIV-1 subtype C, genome deletions were most frequently observed between Integrase and Envelope relative to Gag (p<0.0001–0.001).

e. In 2017, Heiner et. al., in Cell Rep, also described genetic characteristics of the latent HIV-1 reservoir in 3 acute treated and 3 chronic treated male study participants with subtype B HIV. Their data was similar to Bruner et. al. above showing proportions of intact proviruses in participants who initiated therapy during acute/early infection at 6% (94% defective) and chronic infection at 3% (97% defective). In contrast the frequencies in FRESH in acute treated were 52% intact and 48% defective and in chronic infection were 13% intact and 87% defective. These differences could be attributed to the timing of treatment initiation where in the aforementioned study early treatment ranged from 0.6-3.4 months after infection.

(2) Indeed, in the abstract, the authors indicate that treatment was initiated before the peak. The use of the term 'peak' viremia in the hyperacute-treated group could perhaps be replaced with 'highest recorded viral load'. The statistical comparison of this measure in the two groups is perhaps more relevant with regards to viral burden over time or area under the curve viral load as these are previously reported as correlates of reservoir size.

We have edited the manuscript text to describe the term peak viraemia in hyperacute treated participants more clearly (lines 443-444). We have now performed an analysis of area under the curve to compare viral burden in the two study groups and found associations with proviral DNA levels after one year. This has been added to the results section (lines 162-163).

**Reviewer #2:**
(1) Other factors also deserve consideration and include age, and environment (e.g. other comorbidities and coinfections.)

We agree that these factors could play a role however participants in this study were of similar age (18-23), and information on co-morbidities and coinfections are not known.

**Reviewer #3:**
(1) The word reservoir should not be used to describe proviral DNA soon after ART initiation. It is generally agreed upon that there is still HIV DNA from actively infected cells (phase 1 & 2 decay of RNA) during the first 6-12 months of ART. Only after a full year of uninterrupted ART is it really safe to label intact proviral HIV DNA as an approximation of the reservoir. This should be amended throughout.

We agree and where appropriate have amended the use of the word reservoir to only refer to the proviral load after full viral suppression, i.e., undetectable viral load.

(2) All raw, individualized data should be made available for modelers and statisticians. It would be very nice to see the RNA and DNA data presented in a supplementary figure by an individual to get a better grasp of intra-host kinetics.

We will make all relevant data available and accessible to interested parties on request. We have now added a section on data availability (lines 489-491).

(3) The legend of Supplementary Figure 2 should list when samples were taken.

The data in this figure represents an overall analysis of all sequences available for each participant at all time points. This has now been explained more clearly in the figure legend.

**Recommendations for The Authors:**

**Reviewer #1:**
(1) It is recommended that the introduction includes information to set the scene regarding what is currently reported on the composition of the reservoir for those not in the immediate field of study i.e., the reported percentage of defective genomes and in which settings/populations genome intactness has been mapped, as this remains an area of limited information.

We have now included summary of other reported findings in the field in the introduction (lines 89-92, 9498) and discussion (lines 345-350). A more detailed overview has been provided in the response to public reviews.

(2) It may be beneficial to state in the main text of the paper what the purpose of the Raltegravir was and that it was only administered post-suppression. Looking at Table 1, only the hyperacute treatment group received Raltegravir and this could be seen as a confounder as it is an integrase inhibitor. Therefore, this should be explained.

Once Raltegravir became available in South Africa, all new acute infections in the study cohort had an intensified 4-drug regimen that included Raltegravir. A more detailed explanation has now been included in the methods section (lines 435-437).

(3) Can the authors explain why the viral measures at 6 months post-ART are not shown for chronictreated individuals in Figure 1 or reported on in the text?

The 6 months post-ART time point has been added to Figure 1.

(4) Can the authors indicate in the discussion, how the breakdown of proviral composition compares to subtype B as reported in the literature, for example, are the common sites of deletion similar, or is the frequency of hypermutation similar?

Added to discussion (lines 345-350).

(5) Do the numbers above the bars in Figure 3 represent the number of sampled genomes? If so, this should be stated.

Yes, the numbers above the bars represent the number of sampled genomes. This has been added to the Figure 3 legend.

(6) In the section starting on line 141, the introduction implies a comparison with immunological features, yet what is being compared are markers of clinical disease progression rather than immune responses. This should be clarified/corrected.

This has been corrected (line 153).

(7) Line 170 uses the term 'immediately' following infection, however, was this not 1 -3 days after?

We have changed the word “immediately” to “1-3 days post-detection” (line 181).

(8) Can the sampling time-points for the two groups be given for the longitudinal sequencing analysis?

The sequencing time points for each group is depicted in Figure 2.

(9) Line 183 indicates that intact genomes contributed 65% of the total sequence pool, yet it's given as 35% in the paragraph above. Should this be defective genomes?

Yes, this was a typographical error. Now corrected to read “defective genomes” (line 193).

(10) The section on decay kinetics of intact and defective genomes seems to overlap with the section above and would flow better if merged.

Well noted, however we choose to keep these sections separate.

(11) Some references in the text are given in writing instead of numbering.

This has been corrected.

(12) In the clonal expansion results section, can it be indicated between which two time-points expansion was measured?

This analysis was performed with all sequences available for each participant at all time points. We have added this explanation to the respective Figure legend.

**Reviewer #2:**
(1) The statement on line 384 "Our data showed that early ART...preserves innate immune factors" - what innate immune factors are being referred to?

We have removed this statement.

(2) HLA genotyping methods are not included in the Methods section

Now included and referenced (lines 481-483).

(3) Are CD4:CD8 ratios available for the cohorts? This could be another informative clinical parameter to analyse in relation to HIV-1 proviral load after 1 year of ART – as done for the other variables (peak VL, and the CD4 measures).

Yes, CD4:CD8 ratios are available. We performed the recommended analysis but found no associations with HIV-1 proviral load after 1 year of ART. We have added this to the results section (lines 163-164).

(4) Reference formatting: Paragraph starting at line 247 (Contribution of clonal expansion...) - the two references in this paragraph are not cited according to the numbering system as for the rest of the manuscript. The Lui et al, 2020 reference is missing from the reference list - so will change all the numbering throughout.

This has been corrected.

**Reviewer #3:**
(1) To allow comparison to past work. I suggest changing decay using % to half-life. I would also mention the multiple studies looking at total and intact HIV DNA decay rates in the intro.

We do not have enough data points to get a good estimate of the half-life and therefor report decay as percentage per month for the first 6 months.

(2) Line 73: variability is the wrong word as inter-individual variability is remarkably low. I think the authors mean "difference" between intact and total.

We have changed the word variability to difference as suggested.

(3) Line 297: I am personally not convinced that there is data that definitively shows total HIV DNA impacting the pathophysiology of infection. All of this work is deeply confounded by the impact of past viremia. The authors should talk about this in more detail or eliminate this sentence.

We have reworded the statement to read “Total HIV-1 DNA is an important biomarker of clinical outcomes.” (Lines 308-309).

(4) Line 317; There is no target cell limitation for reservoir cells. The vast majority of CD4+ T cells during suppressive ART are uninfected. The mechanism listing the number of reservoir cells is necessarily not target cell limitation.

We agree. The statement this refers to has been reworded as follows: “Considering, that the majority of CD4 T cells remain uninfected it is likely that this does not represent a higher number of target cells, and this warrants further investigation.” (lines 325-326).

(5) Line 322: Some people in the field bristle at the concept of total HIV DNA being part of the reservoir as defective viruses do not contribute to viremia. Please consider rephrasing.

We acknowledge that there are deferring opinions regarding total HIV DNA being part of the reservoir as defective viruses do not contribute to viremia, however defective HIV proviruses may contribute to persistent immune dysfunction and T cell exhaustion that are associated comorbidities and adverse clinical outcomes in people living with HIV. We have explained in the text that total HIV-DNA does not distinguish between replication-competent and -defective viruses that contribute to the viral reservoir.

(6) Line 339: The under-sampling statement is an understatement. The degree of under-sampling is massive and biases estimates of clonality and sensitivity for intact HIV. Please see and consider citing work by Dan Reeves on this subject.

We agree and have cited work by Dan Reeves (line 358).

(7) Line 351: This is not a head-to-head comparison of biphasic decay as the Siliciano group's work (and others) does not start to consider HIV decay until one year after ART. I think it is important to not consider what happens during the first year of ART to be reservoir decay necessarily.

Well noted.

(8) Line 366-371: This section is underwritten. In nearly all PWH studies to date, observed reservoirs are highly clonal.

We agree that observed reservoirs are highly clonal but have not added anything further to this section.

(9) It would be nice to have some background in the intro & discussion about whether there is any a priori reason that clade C reservoirs, or reservoirs in South African women, might differ (or not) from clade B reservoirs observed in different study participants.

We have now added this to the introduction (lines 94-103).

(10) Line 248: This sentence is likely not accurate. It is probable that most of the reservoir is sustained by the proliferation of infected CD4+ T cells. 50% is a low estimate due to under-sampling leading to false singleton samples. Moreover, singletons can also be part of former clones that have contracted, which is a natural outcome for CD4+ T cells responding to antigens &/or exhibiting homeostasis. The data as reported is fine but more complex ecologic methods are needed to truly probe the clonal structure of the reservoir given severe under sampling.

Well noted.